# Heat Shock 70 kDa Protein Cognate 3 of Brown Planthopper Is Required for Survival and Suppresses Immune Response in Plants

**DOI:** 10.3390/insects13030299

**Published:** 2022-03-17

**Authors:** Houhong Yang, Xiaoya Zhang, Hanjing Li, Yuxuan Ye, Zhipeng Li, Xiao Han, Yanru Hu, Chuanxi Zhang, Yanjuan Jiang

**Affiliations:** 1Key Laboratory of Tropical Plant Resources and Sustainable Use, Xishuangbanna Tropical Botanical Garden, Chinese Academy of Sciences, Kunming 650223, China; yanghouhong@xtbg.ac.cn (H.Y.); lizhipeng@xtbg.ac.cn (Z.L.); hanxiao@xtbg.ac.cn (X.H.); huyanru@xtbg.ac.cn (Y.H.); 2University of Chinese Academy of Sciences, Beijing 100049, China; 3Institute of Insect Science, Zhejiang University, Hangzhou 310058, China; 11716078@zju.edu.cn (X.Z.); 21916101@zju.edu.cn (H.L.); yeyuxuan@zju.edu.cn (Y.Y.); 4Center of Economic Botany, Core Botanical Gardens, Chinese Academy of Sciences, Mengla 666303, China; 5Institute of Plant Virology, Ningbo University, Ningbo 315000, China

**Keywords:** brown planthoppers, NlHSC70-3, salivary protein, immune response, rice

## Abstract

**Simple Summary:**

Heat shock 70 kDa protein cognate 3 (NlHSC70-3) of brown planthopper (BPH) is important to the growth and development of BPH. However, if it also regulates plant physiological processes remain unclear. In this study, NlHSC70-3 is identified from a compared protein profile of Nipponbare tissues after BPH infestation. The predicted signal peptide and high levels of transcription in salivary glands and ovaries supported that it is a secreted protein. Function analysis demonstrates that NlHSC70-3 is required for the survival of BPH on rice. Plant immune responses, including flg22-induced ROS bursts and the expression of defense-related genes, are suppressed by NlHSC70-3 in *Nicotiana benthamiana*. Thus, NlHSC70-3 is secreted into rice by BPH and may interfere with plant defense responses to facilitate its survival.

**Abstract:**

The brown planthopper (*Nilaparvata lugens*) is a monophagous pest of rice (*Oryza sativa*), which threatens food security around the world. Insect Heat shock proteins 70 kDa (Hsp70s) play a key role in insect growth and development, however, if they also modulate the plant physiological processes is still unclear. In this study, we identified the Heat shock 70 kDa protein cognate 3 (NlHSC70-3) of BPH from compared protein profiles of Nipponbare tissues after BPH infestation via LC/MS. NlHSC70-3 has a predicted signal peptide and displays high transcription levels in the salivary glands, which further supported that it is secreted into plants by BPH during the feeding process. Using RNA interference (RNAi), we showed that NlHSC70-3 is indispensable for the survival of BPH on rice. Most importantly, NlHSC70-3 mediates the plant immune responses including cell death, flg22-induced ROS burst and defense-related gene expression in *N. benthamiana*. These results demonstrate that NlHSC70-3 may function as an effector manipulating plant physiological processes to facilitate pest survival on rice, which provides a new potential target for future pest control.

## 1. Introduction

The brown planthopper (BPH) is a monophagous insect pest of rice (*Oryza sativa* L.) and damages the host plant by sucking nutrients from the plant stems and leaf tissues through its stylets [1]. In severe outbreaks, wide spread BPH feeding results in a yellowing and wilting phenomenon of rice known as ‘hopper burn’, which can entirely destroy crops [2]. It was estimated that 25 million hectares were infested by BPH between 2005 and 2008, resulting in the loss of 2.7 million tons of rice yields in China [3,4,5]. During feeding, BPH penetrate plant cell walls with their stylets, and then secrete both gelling and watery saliva from their salivary glands into host plants [6]. Watery saliva contains various detoxifying enzymes, proteases and proteins (effectors) that may interact with plant proteins [7,8,9], thus exerting a crucial part in the plant–insect interactions.

In the past decade, proteomics and transcriptomics of the BPH saliva have been extensively studied [6,10,11]. Several BPH salivary proteins, such as NlEG1, NlMLP, NlSEF1 and NlSP1 have been identified and cloned [12,13,14,15]. An endo-β-1,4-glucanase, NlEG1, is delivered into the plant by the planthopper to degrade celluloses in the plant cell walls, which contributes to BPH’s stylets reaching the phloem [12]. NlSEF1 compromises the plant defense response and facilitates to obtain much sap from its host plant by inhibiting calcium signaling that can suppress BPH feeding by callose deposition [13]. A mucin-like protein (NlMLP) is secreted into rice during planthopper feeding [16], which induces a plant immune response, such as cell death, the expression of defense-related genes and callose deposition, which are associated with Ca^2+^ mobilization, the MEK2 MAP kinase and jasmonic acid signaling pathways [14]. Recently, a new effector protein of BPH, NlSP1, has been reported to induce typical immune response including cell death, H_2_O_2_ accumulation, the up-regulation of *PR* genes and callose deposition in rice [15]. These results indicate that BPH salivary proteins play essential roles in crosstalk between insects and their host plants. Therefore, with the progress of research, increasing salivary proteins will be identified to uncover the interaction between the insect and its host plants.

The heat shock response is one of the central mechanisms of biological self-protection in organisms, and this process was first observed in *Drosophila melanogaster* [17]. Insects exhibit resistance to heat stress when stimulated by sublethal temperatures [18,19]. According to the molecular mass, structural characteristics, and functions, HSPs are divided into several types including HSP100, HSP90, HSP70, HSP60, HSP40, and small HSPs [20]. Heat shock 70 kDa proteins (Hsp70s) are the most widely studied proteins in the HSPs family, which participate in an organisms’ response to environmental stresses, such as high/cold temperature, starvation, and anoxia. Subsequently, studies of Hsp70s in insect biology are of great help in understanding the temperature dependence on insect growth and development [21]. The knockdown of *Myzus persicae* heat shock protein 70 (MpHsp70) increases the susceptibility of *M. persicae* to lambda-cyhalothrin (LCT) [22]. In BPH, the expression level of heat shock 70 kDa protein cognate (*NlHsc70*) in macropterous BPH remarkably increased at 32–38 °C, while the expression level of *NlHSC70* in the short-wing type remained unchanged [23]. Additionally, a BPH adult’s response under extreme temperature and pesticide stress may be modulated by expression levels of heat shock protein 70 (HSP70) [24]. Recently, nine HSP70 and 31 DNAJ genes were systematically identified in the BPH genome, and RNA interference (RNAi) revealed that seven NlHSP70s and 10 NlDNAJs play indispensable roles in the nymphal development, oogenesis, and female fertility of *N. lugens* [25]. Most studies have given priority to illustrate the function of Hsp70 in insect growth and development. Nevertheless, whether the Hsp70 protein is secreted into plant cells and the biological roles of Hsp70 protein in rice-insect interactions remain poorly understood.

From protein profiles of the japonica rice (*Oryza sativa*) variety Nipponbare leaf sheath tissues after BPH infestation, we identified the NlHSC70-3 of BPH. NlHSC70-3 has a predicted signal peptide and displays expression levels in the salivary glands, which further supported the hypothesis that it may be secreted along with saliva. RNAi of *NlHSC70-3* results in a low survival rate of BPH feeding on rice plants, which suggested that NlHSC70-3 plays an important role in BPH survival on rice. Moreover, NlHSC70-3 regulates defense response in plants, including induction of cell death, inhibition of flg22-induced reactive oxygen and the expression of defense-related genes. In conclusion, those results demonstrated that NlHSC70-3 may function as a potential effector mediating plant physiological processes to facilitate its survival on rice.

## 2. Materials and Methods

### 2.1. Plant Materials and Insects

The BPH population was collected from Zhejiang University (Jia Xing, China). The BPH was reared with Xiushui 134 rice seedlings in an artificial climate chamber (temperature 27 ± 0.5 °C, relative humidity 50% ± 0.5%, illumination 16 L:8D). *Nicotiana benthamiana* was grown in a climate incubator at Xishuangbanna Tropical Botanical Garden, Chinese Academy of Sciences, Yunnan, China (temperature: 25 ± 1 °C; illumination 16 L: 8D; humidity 70% ± 5%). The japonica rice (*Oryza sativa*) variety Nipponbare was grown in an artificial climate chamber (temperature 27 ± 0.5 °C, relative humidity 50% ± 0.5%, illumination 16 L:8D).

### 2.2. Cloning and Sequencing of NlHSC70-3 Gene

Fifty adult BPHs were ground in liquid nitrogen, and total RNA was extracted by the RNAiso Plus kit (TAKARA, Kusatsu, Japan). The concentration was detected by nucleic acid detector (NanoDrop 2000, Thermo Scientific, Waltham, MA, USA). The cDNA was synthesized using a T7 High yield transcription kit (Vazyme Biotech, Nanjing, China) according to the instructions and stored at −20 °C. Based on the transcriptional library (NCBI Accession No. PRJNA669454, [26]) and the BPH genome libraries in DDBJ/ENA/GenBank, a cDNA sequence was selected as a heat shock protein of the BPH, named NlHSC70-3. The amplification primers designed by Primer 5 are listed in Appendix A. The cDNA template was amplified by PCR with primers. PCR reaction mix consists of Ex-Taq DNA Polymerase 0.5 μL, 10 × Ex Taq Buffer 3 μL, 2.5 mM dNTP 2.5 μL, 2 μL of forward and reverse primers, 2 μL of cDNA template, and 38 μL of ddH_2_O. The amplification procedure was as follows: predenaturation at 94 °C for 5 min, denaturation at 94 °C for 30 s, redenaturation at 60 °C for 30 s, extension at 72 °C for 1.5 min, 35 cycles; extension at 72 °C for 10 min. After the reaction, the molecular weight of PCR products was detected by 1% agarose gel electrophoresis. The amplified product was recovered and linked with pDONR207 and transformed into *Escherichia coli* DH5α (TransGen Biotech, Beijing, China). The bacterial solution (100 μL) was coated on an LB culture plate containing gentamicin (100 μg/mL) and cultured upside down at 37 °C for 12–16 h. The monoclone samples were sent to Shanghai Sangon Sequencing Company for sequencing.

### 2.3. Sequence Analysis of the NlHSC70-3 Gene and Its Encoding Protein

The amino acid sequence was obtained by GenBank translation. BLASTP (http://blast.ncbi.nlm.Nih.gov/blast.cgi, accessed on 15 November 2020) was used to compare the NCBI protein database to further confirm the target genes and obtain homologous sequences. SMART Server (http://smart.embl-heidelberg.De/, accessed on 15 November 2020) was used to analyze conserved domains in coding protein sequences and Signalp-5.0 Server (http://www.cbs.Dtu.dk/services/SignalP/, accessed on 15 November 2020) was used to analyze the signal peptide sequence.

### 2.4. Expression Patterns of NlHSC70-3

RNA extraction and cDNA synthesis of each sample were the same process as previously described. Quantitative primers were designed according to the cDNA sequence of *NlHSC70-3* gene (Appendix A). qRT–PCR amplification was performed using SYBR Green qPCR mix (Vazyme, Nanjing, China) according to the manufacturer’s instructions. The reaction was performed on a quantitative PCR instrument (Roche LightCycler 480). The BPH *18S rRNA* gene (GenBank Accession No. JN662398) was used as an internal reference. Fresh tissues from 100 newly emerged female adult BPH, including salivary glands, guts, intestines, fat bodies, ovaries and testes were isolated and washed at least 3 times in the 1% phosphate buffered saline (PBS) under a stereoscopic microscope. Whole bodies of 100 BPH at different developmental stages from first to fifth instar nymphs, newly emerged female adults, and male adults were collected for RNA extraction. One microgram of each RNA samples was reverse transcribed for qRT-PCR. Each sample has three biological replicates and three technical replicates. The relative expression levels of the *NlHSC70-3* and *18S rRNA* in different BPH tissues and various developmental stages were calculated by the relative quantitative method (∆∆Ct).

### 2.5. dsRNA Synthesis and Injection to Trigger RNAi in BPH

Primers were designed to amplify the 657 bp fragment of *NlHSC70-3* by PCR. The purified nucleotide sequence was cloned into the pMD19-T vector (TAKARA) and the double-stranded RNA was synthesized using a T7 RNAi Transcription Kit (Vazyme, Nanjing, China). A Nanoliter syringe filled with dsRNA was injected into the junctions of the anterior and middle thorax of the third and fifth instar nymphs under a stereoscopic microscope. An amount of 50–250 ng of ds*NlHSC70-3* or ds*GFP* (Green fluorescent protein) was injected by FemtoJet (Eppendorf-Netheler-Heinz, Hamburg, Germany) into each nymph, respectively. The 100 BPH nymphs injected with dsRNA were reared in a plastic tank containing 40–60 Xiushui 134 rice seedlings (three-leaf- stage). To determine the gene silencing efficiency after dsRNA injection, 20 BPH nymphs were randomly picked from the plastic tank, and expression levels of NlHSC70-3 was determined using the procedure described above. The number of surviving BPH nymphs in each plastic tank (diameter, 5.5 cm; height, 20 cm) was recorded daily for 10 days. Six plastic tanks are regarded as a group, of which three plastic tanks contain control BPH and three plastic tanks contain dsHSC70-3 BPH. Experiments in this figure were repeated at least three times with similar trends.

### 2.6. Subcellular Localization of NlHSC70-3 in N. benthamiana and Rice Cells

*NlHSC70-3* was amplified to obtain the target sequence, and part of the encoding signal peptide was deleted. The target sequence was cloned into the pEarleygate101-YFP vector, and the plasmid was transferred into the *Agrobacterium tumefaciens* GV3101 strain. *A. tumefaciens* was co-injected into *N. benthamiana* leaves and allowed to grow in darkness for 48 h. NlHSC70-3-YFP fusion protein and Cyto/Nucleus-RFP marker were co-expressed in rice protoplasts by polyethylene glycol-mediated transformation. Ten micrograms of combined NlHSC70-3-YFP and Cyto/Nucleus-RFP plasmids were added to 100 μL of rice protoplast, then mixed by 110 μL 40% PEG at 28 °C for 15 min. The reaction was terminated by suspending 1.8 mL W5 solution. After stirring with a 450 g horizontal rotor for 3 min, the supernatant was removed with a pipetting device, and the protoplasts were re-suspended in 750 μL W5 solution. Protoplasts were then transferred to 12-well cell culture plates and cultured for 12–16 h under dark conditions at 28 °C. Leaf luminescence was observed with a fluorescence microscope (FV1000; Olympus) to determine the subcellular localization of NlHSC70-3 in *N. benthamiana*.

### 2.7. Defense Gene Expression Analyses of N. benthamiana Leaves

Transient expression of the *NlHSC70-3* in *N. benthamiana* leaves was used to analyze the expression of defense-related genes: salicylic acid (SA)-related marker genes *PR1* (*pathogenesis related gene 1*) and *PR2* and the jasmonic acid (JA)-related marker genes PR3 and PR4 [27,28]. Total RNA was isolated after 48 hr infiltration using the RNAiso Plus kit, according to manufacturer’s instructions, and RT-qPCR was performed as described previously [29]. The primers used for qRT–PCR are listed in Appendix A.

### 2.8. ROS Burst in N. benthamiana

The plasmid was transferred into the *A. tumefaciens* GV3101 strain and injected into *N. benthamiana* leaves (grown in a greenhouse for two and a half weeks). After 48 h of injection, the injected *N. benthamiana* leaves were made into discs with a hole punch (diameter, 4 mm) and put into 96 empty plates (Lumitrac 200, Greiner, no.655075). The pieces were placed in each well individually, and 200 μL ddH_2_O was added. Four replicates were taken for each sample. On the next day, water was replaced with a mock solution containing 34 mg l−1 (*w*/*v*) luminol (Sigma-Aldrich, Shanghai, China) and 20 mg l−1 (*w*/*v*) horseradish peroxidase (Sigma-Aldrich), and a ROS induced solution containing 34 mg l−1 (*w*/*v*) luminol (Sigma-Aldrich), 20 mg l−1 (*w*/*v*) horseradish peroxidase (Sigma-Aldrich) and 100 nM flg22. The luminescence was detected for 1 h with a signal integration time of 2 min using SpectraMax L microplate reader (Molecular Devices, Sunnyvale, CA, USA).

### 2.9. Phylogenetic Tree

Neighbor-joining (NJ) NlHSC70-3, NlHSC70-2, NlHSC70-5, OsHSP70 and OsHSP71.1 of rice, was used to construct a phylogenetic tree by MEGA X software (Version 10.2.6). Species were used in this analysis (GenBank accession numbers are provided in Appendix A).

### 2.10. Data Analysis

Statistical analysis was performed by one-way ANOVA or Student’s *t* test. Graphpad Prism 6.0 was used for making diagrams. All the primers mentioned above are listed in Appendix A.

## 3. Results

### 3.1. Identification of NlHSC70-3 in the Rice Tissue after BPH Infestation

In order to reveal the interaction between the brown planthopper and plants, we successfully obtained a secreted proteins profile of BPH from Nipponbare tissues after BPH infestation through the liquid chromatography-tandem mass spectrometry analyses method. Among these secreted proteins, we identified a protein homologous to the heat shock 70 kDa protein that has a high abundance of peptides with a high score. Compared to the public transcription and protein profiles database, it is heat shock 70 kDa protein cognate 3 (NlHSC70-3, GenBank Accession Number XM_022337451.1). In order to reveal the potential function of it, we cloned the nucleotide sequence of *NlHSC70-3* and further explore its function. The cDNA sequence of NlHSC70-3 contained 1965 bp and encoded 655 amino acids. The signal peptide prediction showed that NlHSC70-3 contained a signal peptide sequence with a length of 21 amino acids (shown as red) (Figure 1 and Appendix A). Using the no-BPH feeding tissues as negative control, we found that 8 amino acid peptides (yellow highlighted) are unique to the NlHSC70-3 (Figure 1), suggesting that NlHSC70-3 is secreted into the host tissues by BPH feeding.

### 3.2. Expression Pattern of NlHSC70-3 in the BPH

Some salivary proteins of insects may be particularly expressed in the salivary glands [30]. To investigate the expression pattern of *NlHSC70-3* in BPH, we analyzed the transcription level of *NlHSC70-3* in different tissues including salivary glands, midgut, fat bodies and ovaries from the adult BPH. Relative expression levels of *NlHSC70-3* in different tissues were normalized against the amount of expression in the gut. According to the results, the expression of *NlSC70-3* genes is consistent with previous results that it displays high expression levels in the ovary [25] (Figure 2A). Interestingly, the abundance of *NlHSC70-3* mRNA was also high in the salivary gland and ovaries, which further supported that *NlHSC70-3* is secreted into plants while feeding and it may also play a role in reproduction.

We also analyzed the expression of *NlHSC70-3* in BPHs at different developmental stages, including the nymphs of the first to fifth instar, female and male adults. The expression of *NlHSC70-3* was increased during the growth stage of BPH nymphs, and reached the highest expression level at the 4th instar nymph stage (Figure 2B). These results demonstrated that *NlHSC70-3* may also play a role in the growth and development of BPH.

### 3.3. Subcellular Localization of NlHSC70-3 in Plant Cells

To further explore the subcellular localization of NlHSC70-3 in plant cells, we deleted the signal peptide of the NlHSC70-3 protein (NlHSC70-3-SP) and fused it with yellow fluorescent protein (YFP). We co-expressed NlHSC70-3-SP:YFP with the cytoplasm/nucleus marker in the *N. benthamiana* leaves and observed that YFP signals were merged with cytoplasm/nucleus signals (Figure 3A) by using the laser scanning confocal microscope, which indicates that NlHSC70-3 localized both in the cytoplasm and nucleus. Then, we prepared rice protoplasts transiently co-expressed NlHSC70-3-SP:YFP with a cytoplasm/nucleus marker to further confirm. Consistent with the *N. benthamiana* results, the YFP signals were merged with cytoplasm/nucleus localization signals (Figure 3B). Together, these results indicate that NlHSC70-3 relevant events may occur both in the cytoplasm and nucleus.

### 3.4. NlHSC70-3 Is Critical to Survival of BPH

According to the transcription levels of *NlHSC70-3* in developmental stages, NlHSC70-3 might also play an essential role in the growth and development of BPH. To elucidate the role of *NlHSC70-3* in different developmental stage, we synthesized double-stranded RNA (dsRNA) from *NlHSC70-3* and injected it into 3rd- and 5th- instar BPH nymphs to reduce the expression of *NlHSC70-3*, and we also inoculated ds*GFP* as the control. As shown in Figure 4A,C, the survival rate of the 3rd instar BPH reached more than 90% even after the 10th day of interference, whereas ds*GFP* injected BPH had no significant dead. However, the survival rate of BPH decreased significantly from the second day after ds*NlHSC70-3* injection in 3rd instar BPH, and below 10% of the BPHs survived at the fifth day of infestation (Figure 4A), which is consistent with Chen’s results that the knockdown of *NlHSC70-3* resulted in a relatively higher mortality rate [25]. qRT–PCR results revealed that the transcript level of ds*NlHSC70-3* was markedly reduced after the RNAi treatments compared to the ds*GFP* (control) treatment in the 3rd instar BPH (Figure 4B). It is worth noting, however, effects of *NlHSC70-3* RNAi in the 5th instar BPH displayed a much weaker mortality rate when compared with the 3rd instar BPH. The survival rate of BPH decreased from the third day, and near 60% of the BPH survived even at the fifth day of infestation (Figure 4C). The results clearly show that *NlHSC70-3* is required for the survival of BPH, and its down regulations had varied effects on these two different developmental stages. As shown in Figure 4D, the transcript level of ds*NlHSC70-3* was significantly reduced after the RNAi treatments compared to the ds*GFP* treatment in the 5th instar nymphs (Figure 4D).

### 3.5. Defense Response Are Inhibited by NlHSC70-3

Since NlHSC70-3 is secreted into rice tissues by BPH (Figure 1) and it is also critical to the survival of BPH (Figure 3), it indicates that NlHSC70-3 may interfere with plant defense responses to facilitate BPH survival. To uncover the potential role of NlHSC70-3 in the host plant, we transiently expressed *NlHSC70-3-SP* (NlHSC70-3 protein without the signal peptide) in *N. benthamiana* leaves and examined the ability of NlHSC70-3 to induce plant cell death by performing *A. tumefaciens*-mediated expression. We set up a concentration (OD600 = 0.1) of NlHSC70-3-SP strains to infect *N. benthamiana* leaves and 35S:YFP was used as a negative control. The results indicated that no cell death was caused by the NlHSC70-3 strains at an early time (Figure 5A). As time goes by, NlSC70-3 led to cell death after 7 days of infection (Figure 5B). To verify whether NlHSC70-3 regulates the plant immune response, we expressed *NlHSC70-3-SP* in *N. benthamiana* to detect the immune response induced by bacterial flagellin22 (Flg22). As shown in the results, flg22-induced ROS bursts were appreciably inhibited by NlHSC70-3-SP compared with the control, YFP (Figure 5C), and flg22-induced ROS bursts were also significantly suppressed by a full-length NlHSC70-3 with the signal peptide, NlHSC70-3+SP (NlHSC70-3 protein with the signal peptide) (Appendix A). To evaluate whether NlHSC70-3 affects other immunity responses, we selected *NbPR1*, *NbPR2*, *NbPR3* and *NbPR4* as resistant marker genes and detected their transcription levels before and after transfecting by NlHSC70-3-SP in *N. benthamiana* leaves. As show in Figure 5D, the transcription levels of *NbPRs* were suppressed by NlHSC70-3 after 24 h inoculation. Collectively, those results demonstrated that NlHSC70-3 is able to suppress plant immune responses, which may be a strategy to facilitate the survival of BPH.

### 3.6. Sequence Analysis of NlHSC70-3 with Rice HSP70s

HSP70s are the most ubiquitous and evolutionarily conserved members among those HSP families [31] and also exist in the rice. Thus, we investigated the evolutionary relationship between OsHSP70s and NlHSC70s, in order to provide clues for further study of plant-insect coevolution. Therefore, we carried out the alignment of the amino acid sequences of NlHSC70s with *Oryza sativa* heat shock protein 70 (OsHSP70, XP_015616495.1) [31,32], and OsHSP71.1 (BAG93163) [32]. Phylogenetic trees of OsHSP70, OsHSP71.1, NlHSC70-2 (KU932402.1) and NlHSC70-5 (XM_022345825.1), which have the high homology with NlHSC70-3 in *N. lugens*, were constructed by MeGa X software using the neighbor-joining (NJ) method. The results show that all the amino acid sequence of NlHSC70-3 and OsHSP70 contained analogous and conserved structural regions of the Hsp70 (Figure 6A). The SMART prediction showed that the amino acid sequence of NlHSC70-3, OsHSP70 and OsHSP71.1 contained conserved structural regions unique to the HSP70 family: GIDLGTTYSVCG, AEAYLGK, IFDLGGGTFDVS and VLVGGSTRIPK domains [25] (Figure 6A). We investigated the degree of similarity between the NlHSC70-3 protein of *N. lugens* and that of other homologous proteins in *N. lugens* or in rice using a BLAST homology search and comparison analysis by NJ. It was showed that NlHSC70-3 was clustered into a single branch distinct from other HSC70s of BPH and rice, indicating NlHSC70-3 may have a different function to others in the HSP70 family in coevolution (Figure 6B).

## 4. Discussion

The Hsp70 family is the most influential and conserved family of heat shock proteins which exist in organisms widely [22,33]. Hsp70s generally produce protective tolerance to stress through molecular chaperone, antioxidant, immune defense and anti-apoptotic effects [34]. Several Hsp70s have been identified in insects, such as *N. lugens* [23,24], *Apis mellifera* [35], and *Liriomyza trifolii* [36]. These studies revealed that Hsp70s may play a pivotal role in insect growth and development, but how Hsp70s functions as potential effectors when they are secreted into the plants is not known. In this study, the full-length cDNA of the *NlHSC70-3* gene was successfully cloned from BPH, and its sequence characteristics were analyzed by the bioinformatics methods. Amino acid sequence analysis revealed that NlHSC70-3 has one conserved ATP-GTP binding site domain and contains characteristic conserved motif regions: GIDLGTTYSCVG, IFDLGGGTFDVS and VLVGGSTRIPK (Figure 1 and Figure 6A). All this information indicates that NlHSC70-3 is a member of the Hsp70 family.

Current studies have found that many Hsp70s of insects do not have predicted signal peptide sequences at the N-terminus, such as two Hsp70s family members, NlHSC70 and NlHSP70 (Appendix A). Interestingly, our Signal prediction showed that NlHSC70-3 has a signal peptide at the N-terminus (Figure 1 and Appendix A), which is the key character of secreted proteins. Most importantly, NlHSC70-3 is secreted into rice tissues by BPH (Figure 1). Additionally, *NlHSC70-3* is expressed in high levels in salivary glands and reproductive tissues (Figure 2A). These evidences demonstrate that NlHSC70-3 is a secreted protein and may play a different role than other typical Hsp70s.

NlHSC70-3 is localized to the cytoplasm and nucleus when expressed in *N. benthamiana* and rice (Figure 3), which is similar to other effectors, Nl12, Nl16, and Nl28 of *N. lugens* [30]. Moreover, *NlHSC70-3* RNAi assay shows the mortality rate of ds*NlHSC70-3* BPH is much higher when compared with the control group when feeding on Nipponbare plants (Figure 4A,C), suggesting that NlHSC70-3 is required for the survival of BPH. In addition, the mortality rate of ds*NlHSC70-3* in the third instar larvae was appreciably higher than that in the fifth instar larvae (Figure 4A,C), implying that NlHSC70-3 may function differently with the developmental stages of BPH.

Due to the fact that salivary proteins and other activators evolved by BPH are able to inhibit the defense response of rice [37,38], we also detected that NlHSC70-3 is involved in mediating plant immune responses. Transient expression analysis showed that NlHSC70-3-SP did not induce cell death in *N. benthamiana* leaves after injected for two days (Figure 5A), so we speculated that NlHSC70-3 may regulate the plant immune response in plants like other effector proteins. As expected, NlHSC70-3 inhibits the flg22-induced ROS burst in *N. benthamiana* leaves (Figure 5C). In addition, NlHSC70-3 substantially suppressed the expression of defense-related genes after 24 h transfection (Figure 5D). Although NlHSC70-3 triggered weak cell death after 7 days (Figure 5B), how NlHSC70-3 is recognized by the R proteins is unclear and need further studies. A HSP70 protein of silkworms regulates the activity of the kinetochore protein Cenp-N activity in the cell cycle by inhibiting the ubiquitin proteasome pathway to control the degradation of centromeres in the cell cycle [39]. Whether NlHSC70-3 has a similar role needs further studies.

## 5. Conclusions

In conclusion, NlHSC70-3 is important to the survival of BPH and may function as an effector protein. From our work, we clarified that NlHSC70-3 is critical to the survival of BPH in different developmental stages. Further research shows that NlHSC70-3 inhibits the flg22-induced ROS burst and defense-related genes expression in plants. Those results indicate that NlHSC70-3 may function as an effector manipulating plant physiological processes to facilitate BPH survival on its host plant.

## Figures and Tables

**Figure 1 insects-13-00299-f001:**
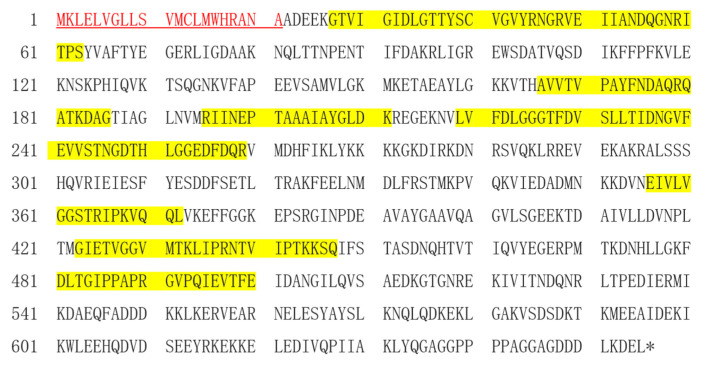
Amino acid sequence of NlHSC70-3. Signal peptide is written in red. The asterisk indicates the stop codon. The highlighted yellow amino acid residues indicate the peptides detected in BPH-infested rice tissue by liquid chromatography-tandem mass spectrometry analyses.

**Figure 2 insects-13-00299-f002:**
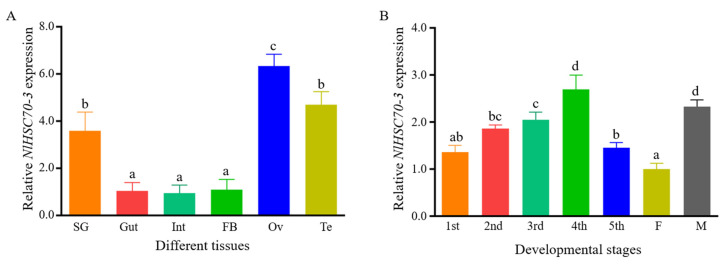
Expression pattern of *NlHSC70-3* in the BPH. (**A**) Relative expression levels of *NlHSC70-3* in different tissues were normalized against the amount of expression in the gut. The relative expression was calculated from three biological replicates. SG = salivary glands, Gut = guts, Int = intestines, FB = fat bodies, Ov = ovary, Te = testis. (**B**) Expression patterns of *NlHSC70-3* gene at developmental stages were normalized against the amount of expression in the female. The 1st to 5th instar; F, female adult; M, male adult, determined by qRT-PCR. Relative expression levels of *NlHSC70-3* in developmental stages were normalized against the amount of expression in the female. Data were analyzed by one-way ANOVA and repeated three times according to three biologically independent samples. Different lowercase letters above bars indicate significant differences.

**Figure 3 insects-13-00299-f003:**
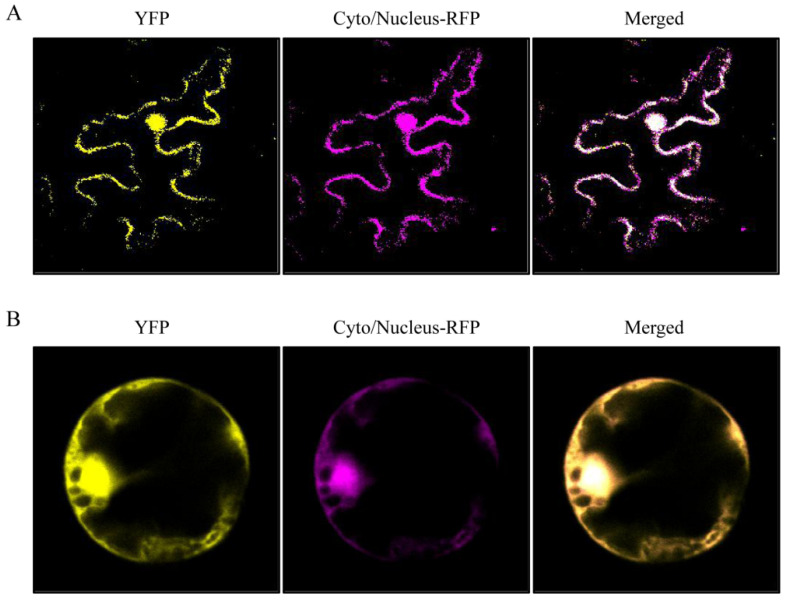
Subcellular localization of NlHSC70-3. (**A**) The NlHSC70-3-YFP fusion protein was expressed in the *N. bethamiana* nucleus and cytoplasm. C-terminal yellow fluorescent protein (YFP) fusion with NlHSC70-3 lacking its endogenous signal peptides. Cyto/Nucleus-RFP marker and NlHSC70-3-YFP fusion protein were co-expressed in *N. bethamiana*. (**B**) Cyto/Nucleus-RFP marker and NlHSC70-3-YFP fusion protein were co-expressed in rice protoplasts by polyethylene glycol-mediated transformation.

**Figure 4 insects-13-00299-f004:**
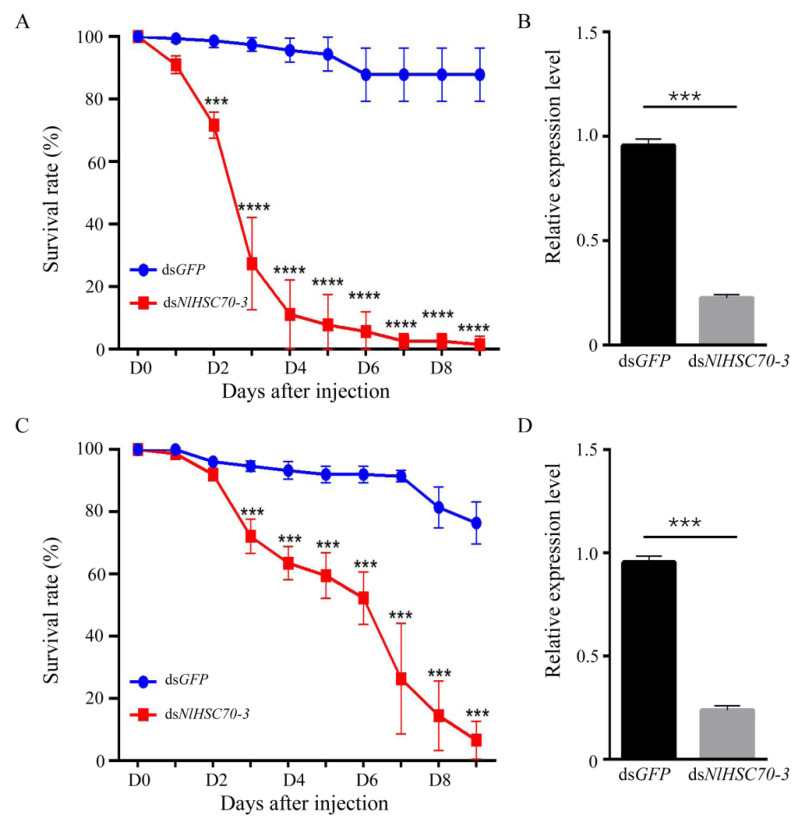
NlHSC70-3 is critical to survival of BPH. (**A**) The survival rate of 3rd instar BPH injected with ds*NlHSC70-3*. (**B**) Transcript levels of *NlHSC70-3* in the control and ds*NlHSC70-3* in 3rd instar nymphs. (**C**) The survival rate of 5th instar BPH injected with ds*NlHSC70-3*. (**D**) Transcript levels of *NlHSC70-3* in the control and ds*NlHSC70-3* in 5th instar nymphs. Data are mean ± s.d. (*n* = 3 biologically independent samples), as analyzed by two-tailed Student’s *t*-test. *p* < 0.001 was considered statistically significant (***), and *p* < 0.0001 was considered statistically significant (****).

**Figure 5 insects-13-00299-f005:**
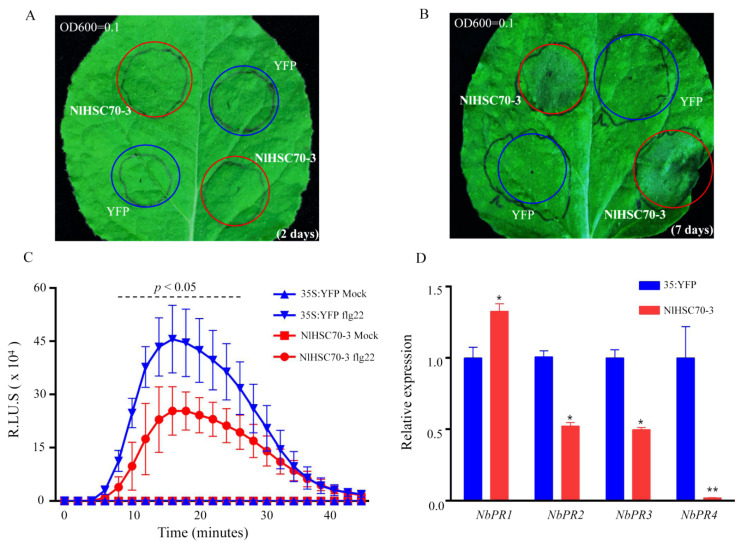
Defense responses are compromised by NlHSC70-3 in *N. benthamiana*. (**A**,**B**) NlHSC70-3 caused cell death at 2 days and 7 days, respectively. Leaves of *N. benthamiana* were infiltrated with *A. tumefaciens* carrying *YFP* and *NlHSC70-3*. (**C**) The flg22-induced ROS burst is suppressed by NlHSC70-3. (**D**) Relative expression of the defense-related genes *NbPR1*, *NbPR2*, *NbPR3* and *NbPR4* inhibited by NlHSC70-3. Data were analyzed by two-tailed Student’s *t*-test. Data are mean ± s.d. (*n* = 3 biologically independent samples). *p* < 0.05 was considered statistically significant (*), and *p* < 0.01 was considered statistically significant (**).

**Figure 6 insects-13-00299-f006:**
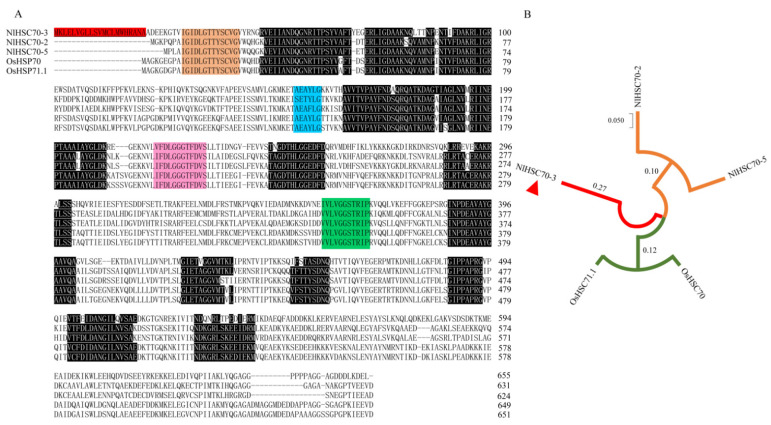
Sequence comparison of NlHSC70-3 with other HSP70s of rice and BPH. (**A**) Alignment of amino acid sequences of NlHSC70-3 with OsHSP70 and OsHSP71.1 of rice, and NlHSC70 and NlHSC70 of BPH. GIDLGTTYSCVG domain, AEAYLGK domain, IFDLGGGTFDVS domain, and VLVGGSTRIPK domain are indicated by orange, blue, pink and green. Similar amino acid sequences of NlHSC70-3, NlHSP70s, OsHSP70 and OsHSP71.1 are indicated by black shading. (**B**) Phylogenetic analysis of heat shock protein 70 from *N. lugens* and rice on amino acid sequences by the neighbor-joining method (1000 replicates). NlHSC70-3 is shown with a red triangle.

## Data Availability

The data presented in this study are available on request from the corresponding author.

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
