# Peer review of "Heat Shock 70 kDa Protein Cognate 3 of Brown Planthopper Is Required for Survival and Suppresses Immune Response in Plants"

_insects, 2022, doi:10.3390/insects13030299_

Round 1

Reviewer 1 Report

In the manuscript “Heat shock 70 kDa protein cognate 3 of brown planthopper is required for survival and suppresses immune response in plants” the authors investigate one of the heat shock proteins of brown planthopper, its expression and its effect on the rice plant physiology. The brown planthopper is a pest of the rice plant and I find the manuscript interesting for the potential readers of the Insect Journal, however, I have some suggestions that the authors need to correct:

  1. Introduction. Considering that the authors investigate rather a long period of time after grasshopper infestation (the leaf necrosis was observed on the 7th day), it would be good to present more information on the feeding behaviour of the insect, e.g. does BHP consume the plant within a short/long time? Does tissue necrosis have any significance later, for example, for the future BPH generation, or it feeds already on the new plants next season?
  2. Methods. Describe which tissues were collected and whether the weight has been recorded. How many females/males were used?
  3. Results. The RNAi experiment shows the effect of the gene on the survival of the BPH 3rd and 5th instars. What is a normal expression of this gene in these developmental stages? Figure 4C is not clear to me (which stage it belongs to?) It also does not help that the labels in Figure4 have non-matching legends (NlHSC70-3 and NlHSP70-3).
  4. Design: The authors have applied the RNAi for the two developmental stages (3rdand 5th instar), at the same time their results show that the gene is expressed in the adult stages in salivary glands and ovaries/testis. Why not to design the RNAi experiment with the adult insect and observe their performance and the plant physiology?
  5. In general, the manuscript is well written but language correction is still needed. For example, for “fed by BPH” I recommend using a better phrase. Also other small corrections are needed in the text.

Reviewer 2 Report

Comments are in attached file. 

Round 2

Reviewer 1 Report

I am reviewing this manuscript for the second round and I should say that the authors fully replied to my comments. I wish the information on the consequences of BPH infestation would be added to the Introduction part as it makes the manuscript even more interesting, but I leave it up to the authors. 

I recommend the manuscript be accepted for publication.